# Pitfalls in AR42J-model of cerulein-induced acute pancreatitis

**Marcus Hollenbach** [ID] * [◉], **Sebastian Sonnenberg** [◉], **Ines Sommerer, Jana Lorenz, Albrecht Hoffmeister**

Medical Department II–Oncology, Gastroenterology, Hepatology, Pulmonology, Infectious Diseases, University of Leipzig Medical Center, D-04103, Leipzig, Germany

◉ These authors contributed equally to this work.
* marcus.hollenbach@medizin.uni-leipzig.de

**Data Availability Statement:** All relevant data are within the paper and its Supporting Information files.

**Funding:** We acknowledge support from the German Research Foundation (DFG) and

## Abstract

### Background

AR42J are immortalized pancreatic adenocarcinoma cells that share similarities with pancreatic acinar cells. AR42J are often used as a cell-culture model of cerulein (CN)-induced acute pancreatitis (AP). Nevertheless, it is controversial how to treat AR42J for reliable induction of AP-like processes. Gene knockout and/or overexpression often remain challenging, as well. In this study, we demonstrate conditions for a reliable induction of proinflammatory markers upon CN treatment in AR42J and high transfection efficacy using Glyoxalase-I (Glo-I) as a target of interest.

### Methods

Effects of dexamethasone (dexa) and CN on cell morphology and amylase secretion were analyzed via ELISA of supernatant. IL-6, TNF-α and NF-κB-p65 were measured via qRT-PCR, ELISA and Western Blot (WB). Transfection efficacy was determined by WB, qRT-PCR and immune fluorescence of pEGFP-N1-Glo-I-Vector and Glo-I-siRNA.

### Results

Treatment of AR42J with 100 nm dexa is mandatory for differentiation to an acinar-cell-like phenotype and amylase production. CN resulted in secretion of amylase but did not influence amylase production. High levels of CN-induced amylase secretion were detected between 3 and 24 hours of incubation. Treatment with LPS alone or in combination with CN did not influence amylase release compared to control or CN. CN treatment resulted in increased TNF-α production but not secretion and did not influence IL-6 mRNA. CN-induced stimulation of NF-κB was found to be highest on protein levels after 6h of incubation. Transient transfection was able to induce overexpression on protein and mRNA levels, with highest effect after 12 to 24 hours. Gene-knockdown was achieved by using 30 pmol of siRNA leading to effective reduction of protein levels after 72 hours. CN did not induce amylase secretion in AR42J cell passages beyond 35.

Universität Leipzig within the program of Open Access Publishing (to MH).

**Competing interests:** The authors have declared that no competing interests exist.

**Abbreviations:** AP, acute pancreatitis; CN, cerulein; Dexa, dexamethasone; FCS, fetal calf serum; Glo-I, Glyoxalase-I; P/S, penicillin and streptomycin; PVDF, polyvinylidene fluoride; qRT-PCR, quantitative real-time polymerase chain reaction; SDS-PAGE, sodium dodecyl sulfate polyacrylamide gel electrophoresis; WB, Western Blot.

## Conclusion

AR42J cells demonstrate a reliable *in-vitro* model of CN-induced AP but specific conditions are mandatory to obtain reproducible data.

## Introduction

Acute pancreatitis (AP) is an inflammatory disease of the pancreas with a wide range of severity that leads to considerable morbidity and mortality [1]. The pathophysiology of the disease is not fully understood yet, but different models are used to examine the pathophysiological processes during the course of AP [2]. An established animal model of AP uses supramaximal doses of cerulein (CN), a cholecystokinin (CCK) analogue. This experimental pancreatitis is characterized by elevation of amylase and lipase levels, cytoplasmatic vacuolization and edema formation [3]. Although this animal model is widely accepted and reveals similar characteristics to human pancreatitis [4], cell culture models of CN-induced AP are important to analyze molecular mechanisms of AP and evaluate innovative therapeutic targets prior to *in-vivo* experiments.

AR42J cells derive from azaserine-induced malignant nodules from the rat pancreas. They maintain the characteristics of normal pancreatic acinar cells including calcium signaling, the synthesis and secretion of digestive enzymes, receptor expression and signal transduction mechanisms [5,6]. Thus, AR42J cells have been widely used to study the function of the exocrine pancreas and as an *in-vitro* model of CN-induced AP [7–12]. However, during the process of immortalization and premalignant transformation, this cell line also exhibits an amphicrine potential that is indicated by neuroendocrine properties, hormone production (mainly gastrin) and autocrine stimulation [13].

AR42J cells need to be stimulated with dexamethasone (dexa) prior to CN treatment to express phospholipase-C-linked receptors (e.g. for CCK) as well as to express and translate secretory proteins. In addition, dexa is necessary to retain the ability of active secretion of digestive enzymes [14]. On the other hand, dexa treatment results in inhibition of cell replication, reorganized endoplasmatic reticulum, increased cell size and protein synthesis [15,16]. Therefore, the design of studies and published methods analyzing CN-induced AP in AR42J cells remain heterogeneous and lead to different published protocols for the implementation of CN treatment in AR42J cells. Indeed, several studies did not mention dexa treatment, although this is essential in inducing the required acinar phenotype. Thus, the aim of this study was to provide evidence for a reliable *in-vitro* model of CN-induced AP in AR42J cells. Furthermore, we analyzed the effect of CN on inflammatory markers and evaluated an optimized protocol for siRNA and plasmid transfection. We used a plasmid containing the sequence of Glyoxalase-I (Glo-I), an enzyme that is involved in oxidative stress and carcinogenesis and that has been analyzed in several projects of our group [17–21].

## Material and methods

### Cell culture

Rat pancreatic AR42J acinar cells were purchased from ATCC (CRL1492) and maintained in DMEM (high glucose, Biochrom/Merck, Berlin, Germany) supplemented with 10% fetal calf serum (FCS, Biochrom/Merck) and 1% penicillin/streptomycin (P/S, PAA, Pasching, Austria). Cells were kept at 37°C with 5% $CO_2$. Medium was replaced every 48 hours and cells were

passaged once a week. Cells were detached by means of trypsin (Biochrom/Merck). If cells were thawed from frozen stocks, they were supplemented with DMEM and 40% FCS. After thawing, cells were allowed to grow and acclimate for 4 to 6 weeks (4 to 6 passages) prior to performing experiments.

## Dexamethasone treatment and cerulein stimulation

Cells were seeded for experiments in 6 well plates (TPP, Sigma-Aldrich, Steinheim, Germany) at concentrations of $6 \times 10^5$/well (for transfection) or $2 \times 10^6$/well (for all other experiments) in serum-free medium. Dexa (Sigma-Aldrich) was added to the medium at concentrations of 100 nM, reaching a final volume of 2 ml, and cells were incubated for 48 hours to allow differentiation to an acinar-like phenotype. Medium was changed after 24 hours.

After dexa pretreatment, medium was changed and 100 nM dexa and/or 10–100 nM CN (Bachem, Bubendorf, Switzerland, final volume 2 ml) was added in serum free medium for 1 and up to 48 hours. Controls received dexa only. For some experiments, cells were incubated with 100 ng/ml LPS (Invitrogen, Thermo Fisher Scientific, Waltham, Massachusetts, USA). Supernatants were collected for amylase secretion assays. RNA isolation was performed by means of RNeasy Mini Kit (74104, Qiagen, Hilden, Germany). Protein lysis buffer (RIPA with complete ultra tablets protease inhibitor high complete, Roche, Mannheim, Germany) was used for protein isolation. Collected samples were stored at -80˚C.

## Measurement of amylase secretion

Supernatants of 6 well plates were thawed and measured via ELISA by the Institute of Laboratory Medicine at the University of Leipzig Medical Center. Results were referred to protein concentrations.

Supernatants were also transferred to sodium dodecyl sulfate polyacrylamide gel electrophoresis (SDS-PAGE). Gels were shaken for 15 minutes at room temperature (RT) in distilled water and stained for 24 hours with Coomassie (Imperial protein stain, Thermo Fisher Scientific) on a shaker. Gels were then destained with distilled water for 2 hours or overnight on a rocking table.

## Western blot analysis

Protein lysates were boiled for 5 minutes at 95˚C in SDS protein buffer (5x laemmli sample buffer, Thermo Fisher Scientific) and separated by SDS-PAGE following transfer to a polyvinylidene fluoride (PVDF) membrane. Primary antibodies were anti-alpha-Amylase (rabbit polyclonal, CST-4017, Cell signaling, Boston, USA), anti-Pancreatic-Lipase-A3 (mouse monoclonal, SC-374612, Santa-Cruz, Texas, USA), anti-Glyoxalase-I (Glo-I, mouse monoclonal, SC-133214, Santa-Cruz), anti-NF-κB (p65 subunit, mouse monoclonal SC-8008, Santa-Cruz) and anti-Vinculin (mouse monoclonal, SC-73614, Santa-Cruz). Secondary antibodies were anti-mouse (goat anti-mouse, 1858413, Pierce / Thermo Fisher Scientific) and anti-rabbit (goat anti-rabbit, 1858415, Pierce). Western Blot signals were quantified using imager (G-Box Chemie XX9, Syngene, Cambridge, UK). Signals were normalized to their respective loading controls using ImageJ-Software (v. 1.48, http://imagej.nih.gov) and: GeneTools (Syngene).

## ELISA

Frozen supernatants or protein lysates were thawed on ice. For TNF-α-ELISA (BD Rat TNF ELISA, 560479, Becton Dickinson, New Jersey, USA) measurements, 20 µl of supernatant or

protein lysate were used according to the instructions from the manufacturer. Reagents were prepared as indicated in the manual. Strips were inserted into the ELISA plate depending on the necessary well count. Assay diluent and standards or samples were pipetted into the corresponding wells, shaked and incubated for 2 hours at RT. Supernatants were removed, the wells washed five times and all liquids properly removed. Detection antibody was added and the wells incubated for 1 hour at RT followed by another washing step. Enzyme working reagent was added and incubated for 30 minutes at RT. Wells were washed 7 times followed by pipetting of One-Step-Substrate reagent and incubated for 30 minutes at RT in the dark. Stopping solution was added and absorbance was measured at 450nm within 30 minutes. Results were referred to protein concentrations.

## qRT-PCR

RNA was isolated with RNeasy Mini Kit (Qiagen) following the instructions of the manufacturer. Quantitative real-time PCR (qRT-PCR) was performed by means of QuantiTect SYBR Green RT PCR Kit (one-step PCR, Qiagen 204243) according to the manual. Reverse transcription was performed at 50˚C for 30 minutes followed by PCR-program (95˚C for 20 minutes, 40 x (94˚C for 15 seconds, 55˚C 20 seconds, 72˚C 20 seconds). A melting curve analysis was performed within any procedure. The following QuantiTect Primer Assays (Qiagen) were used: IL-6 (QT00182896), TNF-$\alpha$ (QT02488178) and beta-actin (QT00193473). Experiments were performed on a Light Cycler 3.5 (Roche), results were calculated by Relative Expression Software Tool (REST®, Qiagen).

## Glo-I plasmid generation

Total RNA was isolated from AR42J cells using the RNeays Mini Kit (Qiagen) following the instructions from the manufacturer. First-strand cDNA was generated from normalized RNA amounts using Oligo-(dT)-primers and the Omnisript RT Kit (205111, Qiagen) according to the instruction manual. Glo-I insert with Bgl-II and Eco-RI cutting sites was constructed with Glo-I primers (5'-3' (ffw): `GACAGATCTATGGCAGAGCCACAGCCA`, 3'-5' (rev): `CAGGAATT CCTAAATAATTGTTGCCATTTTGTT`) and the following PCR-program (Taq-polymerase, 201203, Qiagen): 94˚C for 3 minutes, 35 x (94˚C for 1 minute, 52˚C for 1 minute, 72˚C for 1 minute), 72˚C for 10 minutes and finally 4˚C until the removal of probes. PCR products were mounted on gel electrophoresis and corresponding lanes were cut and purified. The inserts and a pEGFP-N1 vector (6085–1, Clontech, California, USA) were digested by Bgl-II (R0144S, NEB, Massachusetts, USA) and EcoRI (R0101S, NEB) and the vector was dephosphorylated (Shrimp alkaline phosphatase, EF0511, Fermentas / Thermo Fisher Scientific). After ligation (Rapid DNA ligation kit, K1412, Fermentas), E. coli (Dh5 Alpha, Thermo Fisher Scientific) were transformed with the pEGFP-N1-Glo-I vector. For this purpose, E. coli were heat shocked for 45 seconds at 42˚C, incubated for 2 minutes on ice and shaken for 1 hour at 37˚C. Then, E. coli were plated on LB agar plates with canamycin resistance and incubated overnight at 37˚C and 5% $CO_2$. After an incubation of 24 hours, colonies were picked and sequenced by means of CMV primer (fw: `GTCAATGGGAGTTTGTTTTGG`; Sigma-Aldrich). Midi preparation (Qiagen Plasmid Midi Kit, K12144, Qiagen) was conducted from colonies with correct vector sequence.

## Transfection

Cells were seeded in 6 well plates at concentrations of 6 x $10^5$/well in serum-free medium containing 100 nM dexa and incubated for 48 hours with a medium change after 24 hours. For transfection, two vials were prepared at RT: vial A containing 2.5 to 5 μg plasmid DNA

(pEGFP-N1-Glo-I or pEGFP-N1 control) in a volume of about 100–125 μl. Serum-free medium was added to reach a final volume of 150 μl. Vial B included 145 μl serum-free medium and 5 μl transfection reagent (Lipofectamin2000, Invitrogen, Thermo Fisher Scientific). Vials were gently mixed by means of up and down pipetting and incubated for 5 minutes at RT. Consecutively, the content of vial A was transferred to vial B, mixed again and incubated for 10 to 15 minutes at RT. Then, 250 μl of mixed solution (vial A + B) were transferred to the corresponding well by slow dropping. The 6 well plates were incubated on a shaker (50 rpm) for 20 minutes at 37˚C with 5% $CO_2$. Finally, well plates were incubated for 6 to 24 hours at 37˚C and 5% $CO_2$ in the incubator. Cells were harvested for protein and mRNA analysis as described above. Transfection reagent without plasmid DNA was used as a sham control.

For siRNA transfection, Lipofectamin RNAi Max (13778–150, Thermo Fisher Scientific), 30–300 pmol Glo-I-siRNA (Silencer Pre. Designed siRNA Rat Glyoxalase 1, 201921, Thermo Fisher Scientific) in 2 ml volume (6 well cavity) and control-siRNA (Silencer negative control 1 siRNA, AM4611, Applied Biosystems / Thermo Fisher Scientific) were used.

## Statistical analysis

Results are expressed as mean ± standard deviation (S.D.). For comparison of only two groups, the Student's test was performed. For three or more groups, the one-way ANOVA with Bonferroni post-test was used. P values <0.05 were considered statistically significant. All experiments represent means of at least three independent experiments. GraphPad Prism 4.0 software was used for calculation and drawing of graphs.

## Results

### Cell culture and passage

Several modifications to maintain AR42J cells were reported including Ham's F12 [22], F12K [23], Dulbecco's modified eagle medium (DMEM) [14], minimal essential medium (MEM) [9] and RPMI1640 medium [24]. We used the above mentioned cell culture conditions (DMEM with 10% FCS at 37˚C with 5% $CO_2$). Under these conditions, cells were stable and were passaged once a week. Incubation of cells with 20% FCS did not show any advantage with regard to cell morphology or proliferation. We did not add dexa during maintenance of cell culture but instead for experiments to ensure differentiation to an acinar-like phenotype. Cells were used up to passage 35. AR42J cells of higher passages showed modified properties. Cells grew more rapidly, were not able to differentiate through the use of dexa and did not release amylase (see below). Also, morphological changes appeared. The cells with a higher number of passages were more round and thicker.

### Dexamethasone pretreatment and cerulein stimulation

Treatment with 100 nM dexa was mandatory for differentiation of AR42J to an acinar-like phenotype. Compared to controls cell morphology was rounder and thicker with numerous granular deposits after treatment with dexa (Fig 1). Co-treatment of cells with dexa and CN resulted in similar morphologic alterations as dexa alone. Dexa treatment led to a significant increase of amylase and lipase content in AR42J cells (more than fourfold, Fig 2E1 and 2E2) but only slightly induced secretion of amylase into cell supernatant (Fig 2A1–2B).

CN treatment of cells without dexa preincubation induced neither an increase in amylase storage content nor amylase secretion to supernatant. These results were reproducible with low (10 nM) or high (100 nM) concentrations of CN (Fig 2A1–2B, 2E1 and 2E2). For the following experiments we used 100nM CN as this concentration of CN induced highest amounts

**Control**

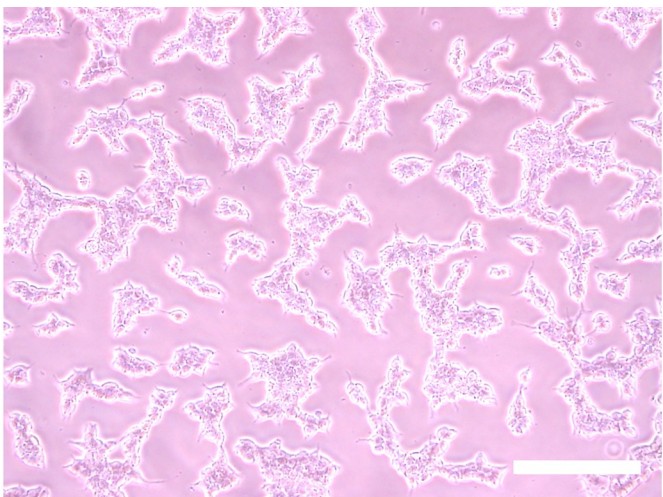

**Dexa 100 nM**

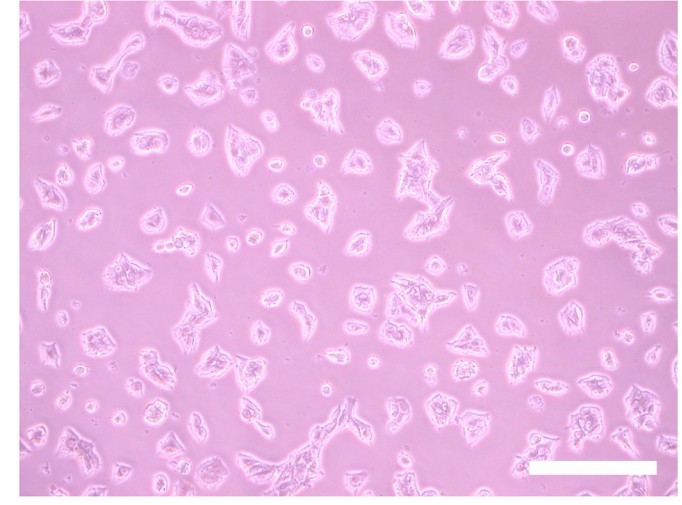

**Dexa + CN 100 nM**

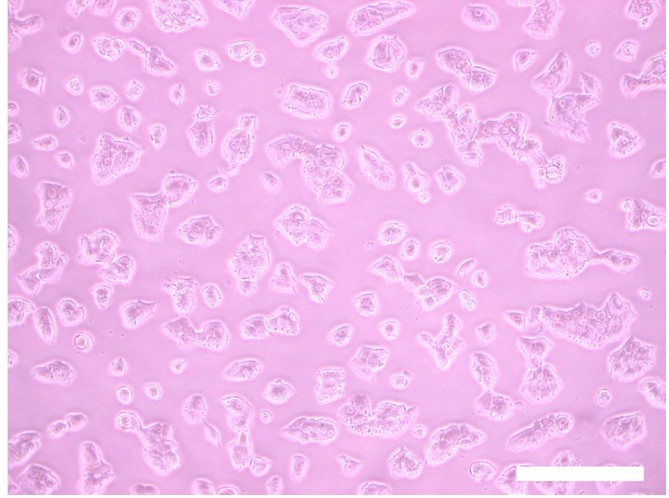

**Fig 1. Effect of dexamethasone and ceruleine on AR42J morphology.** Cells were incubated under standard conditions (DMEM supplemented with 10% fecal calf serum and 1% penicillin/streptomycin, upper image) with 48 hours of dexamethasone (dexa, middle) or 48 hours of dexa followed by 24 hours of cerulein (CN, lower image). Dexa led to a differentiation in an acinar like phenotype. Cell morphology was rounder and thicker with numerous granular vessels inside. Co-treatment of dexamethasone and CN resulted in similar morphologic alterations as did dexamethasone alone. Dexa: dexamethasone, CN: cerulein. Scale bars: 100μm.

of amylase release. After 48 hours of dexa pre-treatment, CN was able to significantly increase amylase secretion of AR42J into cell supernatant (more than twofold in Coomassie-staining and more than fourfold in ELISA measurements, Fig 2A1–2B). CN-induced amylase secretion was measured after incubation of at least 1 hour and up to 24 hours. After more than 24 hours, the effect of CN on amylase secretion was no longer detectable (Fig 2D). In contrast, addition of 100 ng/ml LPS alone was not able to induce amylase release from AR42J cells. Moreover, the co-treatment of LPS and CN showed no effect compared to CN without LPS (Fig 2C).

## Effect of dexamethasone and cerulein on TNF-α, IL-6 and NF-κB

Inflammatory markers are the subject of numerous reports that analyzed AP *in-vitro*. Recently published studies mainly examined TNF-α, IL-6 and NF-κB by various methods in cell supernatant and cytosol (Table 1). However, these studies have, at least in part, contradictory results. Thus, we evaluated gene expression, protein production and secretion upon CN stimulation of different inflammatory parameters.

CN-treatment for 24 hours showed no induction of TNF-α on mRNA levels (Table 2). Comparable results were measured for IL-6 qRT-PCR: treatment of AR42J for 24 hours with 100 nM CN did not result in induced IL-6 transcription (Table 2). Next, TNF-α secretion was examined by ELISA in AR42J supernatants. We were not able to detect any increase of TNF-α secretion, neither upon CN stimulation nor dexa treatment. The absence of TNF-α secretion was independent of incubation time (Fig 3A1). In contrast, CN-induced TNF-α production was indicated by ELISA measurement of protein lysates. The gain in TNF-α concentration was significant after 6 hours of incubation and reached a 1.5-fold increase after 24 hours (p<0.001, Fig 3A2).

In addition, the effect of CN on NF-κB p65 subunit was analyzed. Dexa treatment without CN did not influence protein expression of NF-κB. However, co-treatment of dexa and CN increased the protein expression of NF-κB 2.7-fold after 24 hours of incubation (p<0.001, Fig 3B1 and 3B2).

## Transient transfection of AR42J cells

Transfection of AR42J cells was previously described [22] but remains challenging in daily practice. Thus, we evaluated a modified protocol (see methods section) for optimized transfection. We used the GFP-tagged vector pEGFP-N1-Glo-I containing the sequence for the enzyme Glo-I that was used in several projects [21] in our lab and commercial Glo-I-siRNA. Verification of effective transfection was performed via WB and qRT-PCR at different incubation periods. Optical validation of transfection was conducted by means of the fluorescent GFP (Fig 4A1–4A3). Highest efficacy of transient transfection was detected after a treatment of 12 hours (> fourfold increase WB, 1.5-fold in qRT-PCR, p<0.001). After an incubation of 6 or 24 hours, protein expression was lower compared to 12 hours but also significantly elevated in relation to controls (p<0.05). Results were also confirmed by qRT-PCR analysis (all Fig 4B1–4C). In order to evaluate if higher plasmid concentrations would result in improved transfection efficacy, we also used doubled plasmid amounts (5 μg instead of 2.5 μg in 300 μl

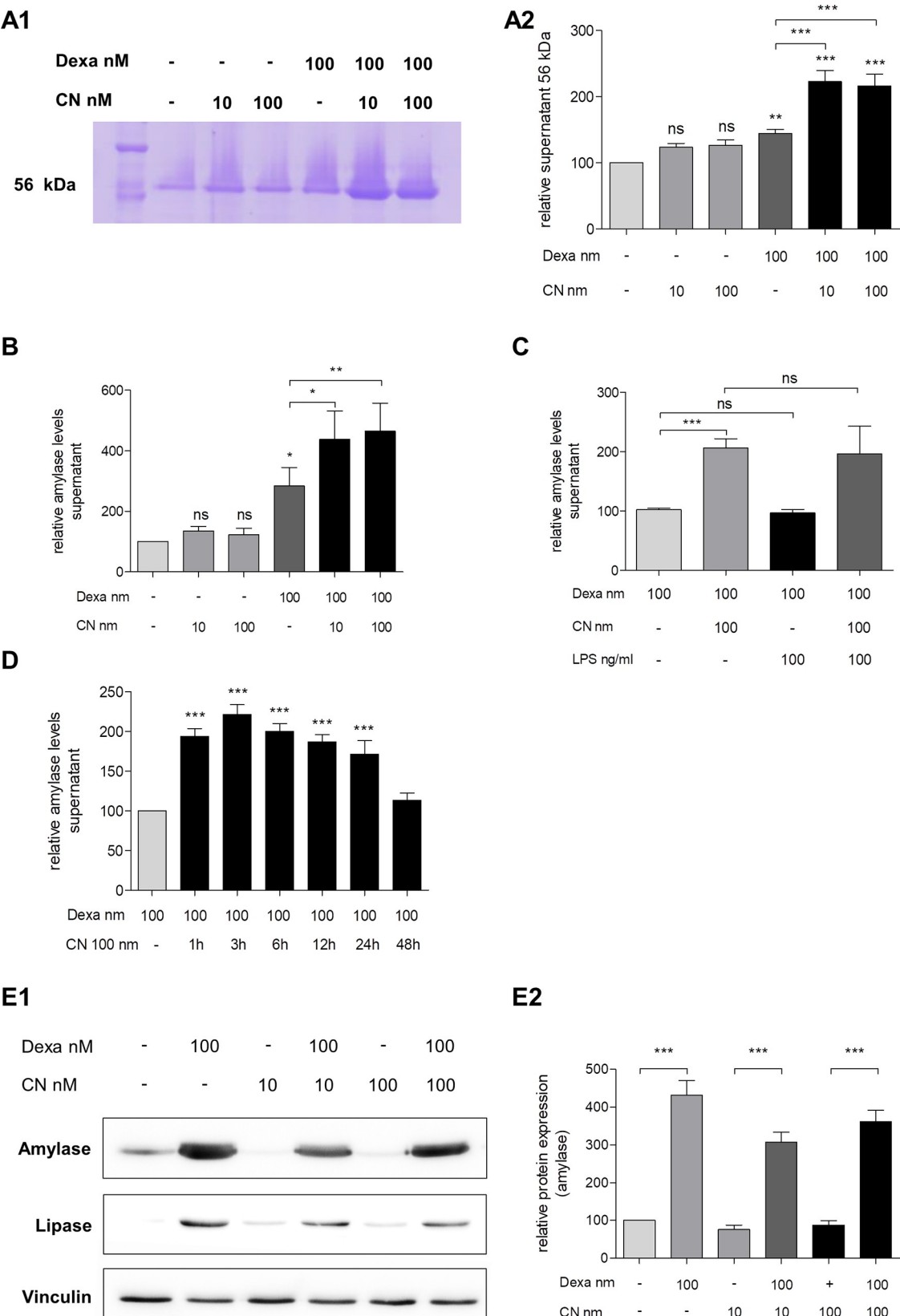

**Fig 2. Effect of dexamethasone and cerulein on secretion of AR42J cells. A1-A2**, analysis of amylase secretion of AR42J following dexamethasone (dexa) and cerulein (CN) treatment by SDS-PAGE gel electrophoresis and Coomassie staining. Representative image

of gel electrophoresis (**A1**) indicated stimulation of amylase secretion (56 kDa lane) after CN stimulation. Treatment of dexa alone resulted in slight increase of amylase secretion but not CN stimulation without prior dexa treatment. Quantifications (**A2**) confirmed the CN-induced amylase secretion after dexa pre-treatment. The preliminary results of the SDS-PAGE gel electrophoresis were confirmed by ELISA of AR42J supernatant. Co-treatment of dexa and CN but not CN alone resulted in dramatically elevated amylase secretion (**B**). The highest effect of the CN stimulation after dexa pretreatment was measured after 3 hours of incubation but also confirmed up to 24 hours (**D**). LPS was not able to induce a secretion of amylase by AR42J and did not show any effect in co-treatment of LPS and CN (**C**). **E1-E2**, Western Blot analysis indicated that dexa but not CN is mandatory for the production of amylase and lipase in AR42J. Representative images show no influence of CN on enzyme production (**E1**), quantifications (**E2**) confirmed the necessity of dexa for amylase production. CN: cerulein. Dexa: dexamethasone. Results are expressed as mean ± S.D. of at least three independent experiments. * $P < 0.05$, ** $P < 0.01$, *** $P < 0.001$.

**Table 1. Published studies and protocols of cerulein treatment in AR42J cells.**

| Study | Dexa c (nM) | Dexa time (h) | CN c (nM) | CN time (h) | LPS | NF-kB | TNF-a | IL-6 | Amylase secretion |
|---|---|---|---|---|---|---|---|---|---|
| Bhatia 2012 | NA | NA | 1–100 | 0–6 | no | No | no | no | yes |
| Cai 2016a | NA | NA | 100 | 24 | no | p-p65 (WB) | ELISA (S) | ELISA (S) | no |
| Cai 2016b | NA | NA | 10 | 0–24 | no | p-p65 (WB) | ELISA (S) | ELISA (S) | no |
| Chan 2011 | NA | NA | 100 | 1–4 | no | no | no | no | yes |
| Chen 2010 | NA | NA | 10 | 24 | no | no | WB, qPCR | WB, qPCR | no |
| Chen 2011 | NA | NA | 10 | 0–48 | no | p65 (WB) | WB, qPCR | WB, qPCR | no |
| Chen 2018 | NA | NA | 100 | NA | no | p-p65 (WB) | ELISA (L) | ELISA (L) | No |
| Chu 2013 | NA | NA | 10 | 2–10 | no | no | no | no | no |
| DeLisle 2005 | 100 | 48 | 100 | 0.66 | no | no | no | no | yes |
| Eum 2003 | 10–50 | 48 | no | no | no | no | no | no | yes |
| Gu 2018 | NA | NA | 10 | 0–24 | no | No | ELISA (S) | ELISA (S) | no |
| Guthrie 1991 | 100 | 0–144 | no | no | no | no | no | no | yes |
| Huang 2012 | NA | NA | 10 | 12 | no | no | WB ELISA | no | yes |
| Huang 2012 | NA | NA | 10 | 0–48 | no | no | no | no | no |
| Jiang 2017 | NA | NA | 100 | 2–24 | no | p65 (WB) | qPCR | qPCR | yes |
| Ju 2011 | NA | NA | 10 | 0–24 | no | no | no | no | no |
| Ju 2017 | NA | NA | 10 | 0–4 | no | no | no | No | no |
| Kandil 2006 | 100 | 48 | no | no | no | no | no | no | no |
| Kimura 1998 | 25–100 | 18 | no | no | no | no | no | no | no |
| Kullman 1996 | 50 | 48 | no | no | no | no | no | no | no |
| Lee 2003 | NA | NA | 10–1000 | 0–24 | no | no | no | ELISA (S) | no |
| Lee 2007 | NA | NA | 10 | 4 | no | no | no | qPCR | no |
| Lim 2008 | NA | NA | 1 | NA | no | no | no | no | no |
| Liu 2014 | NA | NA | 10 | 4–24 | 10 mg/l | no | ELISA (S) | ELISA (S) | yes |
| Liu 2017 | NA | NA | NA | 24 | Yes | no | no | no | no |
| Logsdon 1985 | 10 | 48 | 0.01–1000 | 0.6 | no | no | no | no | yes |
| Logsdon 1987 | 100 | 0–72 | no | no | no | no | no | no | yes |
| Nakamura 2010 | NA | NA | 100 | 3 | no | no | no | no | no |
| Sledzinski 2013 | NA | NA | 10 | 0–4 | no | no | no | no | no |
| Song 2017 | NA | NA | 10 | 1–24 | No | no | no | qPCR, ELISA (S) | no |
| Tang 2017 | NA | NA | 10–1000 | 24 | no | p65 (WB/PCR) | no | ELISA (S) | yes |
| Wan 2008 | NA | NA | 10 | 0–24 | no | p65, p50 (WB) | no | no | no |
| Wang 2015 | NA | NA | 10 | 0.5–48 | no | no | ELISA (L) | ELISA (L) | No |
| Wang 2018a | NA | NA | 10 | 0.25–24 | no | p65 (WB) | qPCR | qPCR | no |
| Wang 2018b | NA | NA | 10 | 24 | no | no | no | no | no |
| Wu 2014 | 100 | 48 | NA | NA | NA | no | no | no | yes |
| Xie 2017 | NA | NA | 10 | 24 | no | p-p65 (WB) | ELISA (S) | ELISA (S) | no |
| Xue 2009 | NA | NA | 10 | 24 | no | no | no | no | yes |
| Yu 2005a | NA | NA | 0.1–100 | 0–24 | no | EMSA | no | qPCR, ELISA (S) | np |
| Yu 2005b | NA | NA | 10 | 1 | no | no | no | no | no |
| Yu 2006 | NA | NA | 10 | 1–24 | no | no | no | no | no |
| Yu 2007 | NA | NA | 10 | 0.24–24 | no | no | no | no | no |
| Yuan 2008 | NA | NA | 10–100 | 0.17 | no | EMSA | no | no | no |
| Zhang 2015 | NA | NA | NA | NA | no | no | no | no | no |
| Zhao 2018 | NA | NA | 10 | 16 | no | no | no | No | yes |
| Zhao 2018 | NA | NA | 10 | 0–12 | 10 mg/l | WB | no | no | no |
| Zhou 2016 | NA | NA | 10 | 24 | no | no | qPCR | qPCR | yes |

**Table 2. Detection of IL-6 and TNF-α in AR42J cells.**

| Parameter | Dexa 100 nm | Dexa 100 nm + CN 100 nm | p |
|---|---|---|---|
| IL-6 | 36.1 ± 2.4 | 36.3 ± 1.5 | 1.0 |
| TNF-α | 25.8 ± 0.7 | 26.7 ± 0.7 | 0.2 |
| beta-actin | 18.0 ± 0.5 | 17.4 + 0.3 | 0.4 |

transfection reagent). Results indicated a deteriorated protein expression upon high volumes of the used Glo-I plasmid (Fig 4D1 and 4D2).

Next, we analyzed the efficacy of Glo-I knockdown by siRNA under certain circumstances. Expression of Glo-I on protein level was effectively reduced. This reduction was incipient after

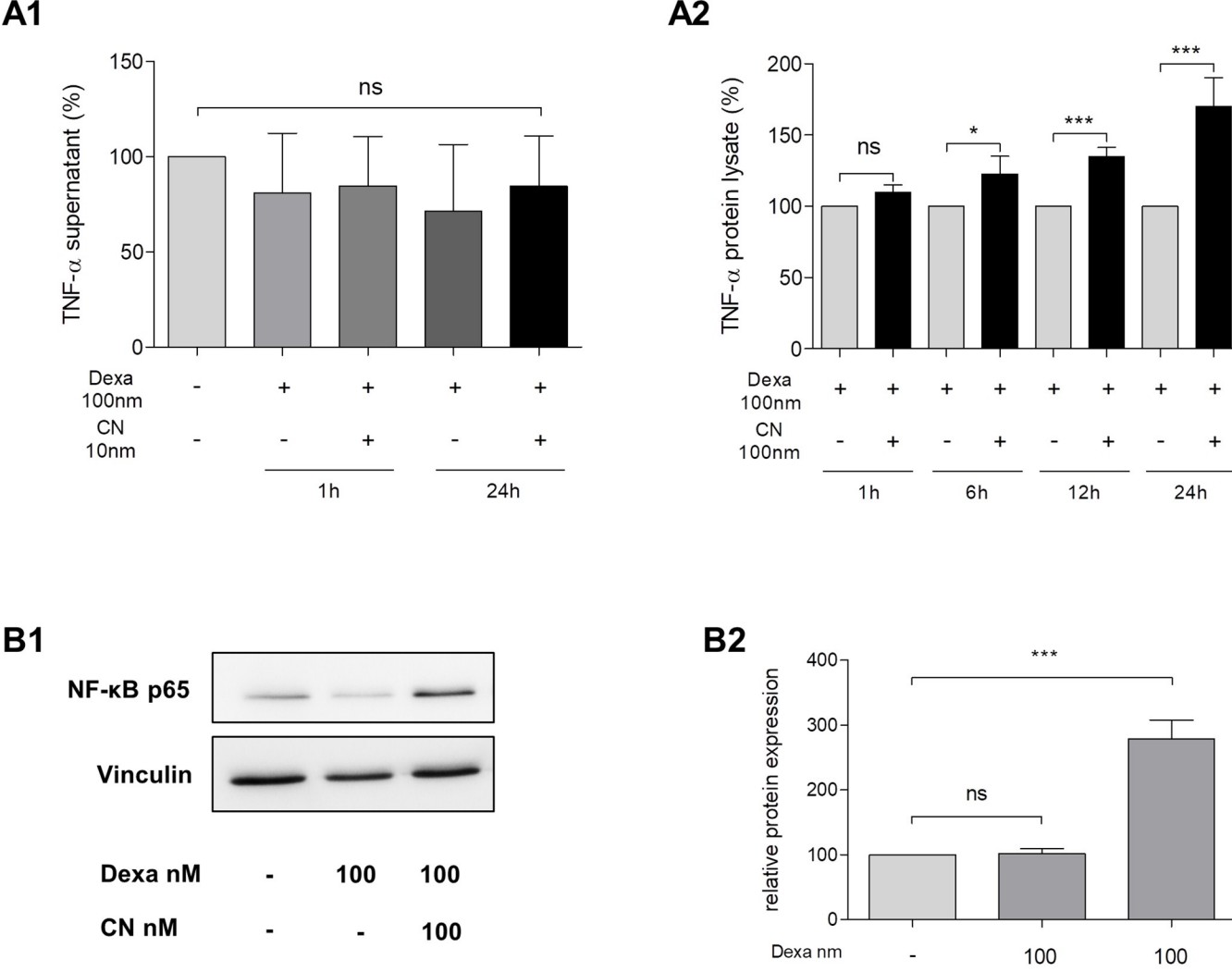

**Fig 3. Effect of dexamethasone and cerulein on TNF-α and NF-κB. A1-A2**, effect of dexamethasone (dexa) and cerulein (CN) on TNF-α. Cells were treated for 48 hours with 100 nM dexa (except controls in (**A1**)), followed by stimulation of 100 nM CN for varying times. TNF-α concentrations were determined by ELISA in supernatant (**A1**) or protein lysate (**A2**). Neither the use of dexa nor CN resulted in alterations of TNF-α release. However CN resulted in significant elevation of TNF-α production (**A2**). **B1-B2**, effect of dexa and CN on NF-κB. Dexa did not influence p65 unit of NF-κB on protein levels but CN significantly induced its expression 2.5 fold. Representative Western Blot images are shown in **B1**, quantifications (**B2**) confirmed significant induction by CN. CN: cerulein, Dexa: dexamethasone. Results are expressed as mean ± S.D. of at least three independent experiments. *** $P < 0.001$.

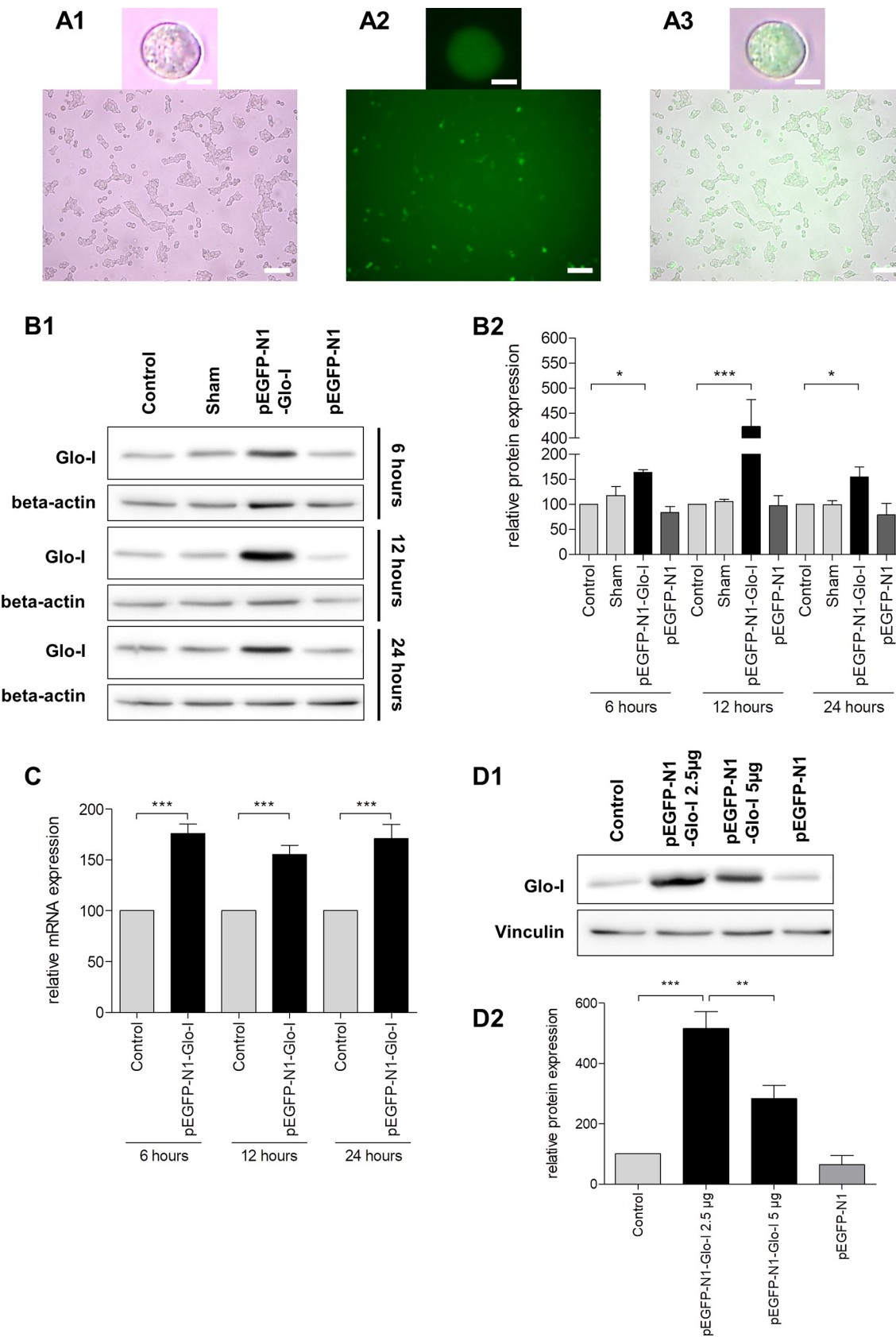

**Fig 4. Transient transfection of AR42J cells. A1-A3**, transfection of AR42J with pEGFP-N1-Glo-I vector. Cells were seeded and incubated with 100 nM dexamethasone for 24 hours. Representative images at 20x magnification (upper line) and 5x magnification (lower line) indicate transfection efficacy by green colored cells in FITC-fluorescence channel **(A2)** and overlay **(A3)** compared to white light images **(A1)**. **B1-D2**, transfection efficacy under different conditions, analyzed by Western Blot (WB) and qRT-PCR. Representative WB images are shown in **B1**. Elevated protein expression of Glyoxalase (Glo-I) was detectable after 6h and highest after 12 hours of transfection. Quantification **(B2)** of at least three independent experiments confirmed this increase to be significant. Controls, sham-treatment and the use of pEGFP-N1 vector confirmed the specific effect of the plasmid transfection. Higher amounts of plasmid concentrations deteriorated the expression of the target gene **(D1-D2)**. Significant elevation of Glo-I expression was also confirmed on mRNA-levels by qRT-PCR **(C)**. Glo-I: Glyoxalase-I. WB: Western Blot. Results are expressed as mean ± S.D. of at least three different experiments. Scale bars: 10µm (upper line), 100µm (lower line). * P<0.05, ** P<0.01, *** P<0.001.

24 hours of incubation (reduction to 83%, p<0.05) and highest after 72 hours (reduction to 51%, p<0.001, Fig 5A1 and 5A2) compared to controls and sham transfection. Similar results were obtained in the analysis on mRNA levels. Glo-I expression was reduced to 11% in qRT-PCR after 24 hours of treatment (p<0.001, Fig 5B1). Interestingly, higher amounts of siRNA (100 pmol and 300 pmol) did not decrease the Glo-I expression on mRNA-levels further compared to the lower concentration of 30 pmol (Fig 5B2). These results were also confirmed by using different amounts of transfection reagent (no significant difference in using 6µl or 12µl of reagent, Fig 5B3). Additionally, recent studies in AR42J cells showed a potential benefit in using a shaker for transfection [25]. Interestingly, we found no difference in Glo-I downregulation in cells that were shaken for 24 hours compared with stationary cells indicated by qRT-PCR (Fig 5B2).

## Effect of cell passage on experimental setting

Our observations during cell culture maintenance indicated that AR42J cells show distinct characteristics in higher cell passages with regard to proliferation and morphology. Thus, we analyzed if cells differ in their capability to release amylase after several passages. AR42J, that had fewer than 35 passages, showed reliable amylase release upon CN stimulation. In contrast, in experiments performed with AR42J beyond passage 45, we were not able to detect any increase in amylase release (Fig 6A). Interestingly, AR42J cells that were of passages higher than 35, revealed an increased transfection efficacy. In these cells transfection with Glo-I-siRNA resulted in significantly more reduction of gene expression on mRNA levels (Fig 6B).

## Discussion

AR42J cells are an immortalized rat pancreatic cell line that share similarities with acinar cells, including synthesis and secretion of digestive enzymes, as well as receptor expression and signal transduction mechanisms [5,6]. Therefore, AR42J cells demonstrate an appropriate *in-vitro* model for the study of the function of the exocrine pancreas and experimental AP induced by CN [7–12]. However, as indicated by Table 1, the conditions and used protocols for the analysis of CN-induced pancreatitis in AR42J remain heterogeneous and are, at least in part, contradictory.

First, different conditions have been reported to maintain AR42J in cell culture. Cells were cultivated in Ham's F12 [22,26,27] and F12K [11,12,23,28–39] medium, Dulbecco's modified eagle medium (DMEM) [7,10,14,40–58], minimal essential medium (MEM) [9,59–61], RPMI1640 [24] or not stated [62–64]. Medium was supplemented with 10% and up to 20% FCS in all papers, and cells were cultivated at 37˚C with 5% CO2. Thus, we decided to use DMEM supplemented with 10% FCS in our experimental setting, as the majority of publications used this medium for AR42J culture. These conditions resulted in stable cell cultures that were passaged once a week. The increase of FCS supplementation to 20% FCS did not show any superiority compared to 10%. We only used cells up to passage 35. This is important, as

**A1**

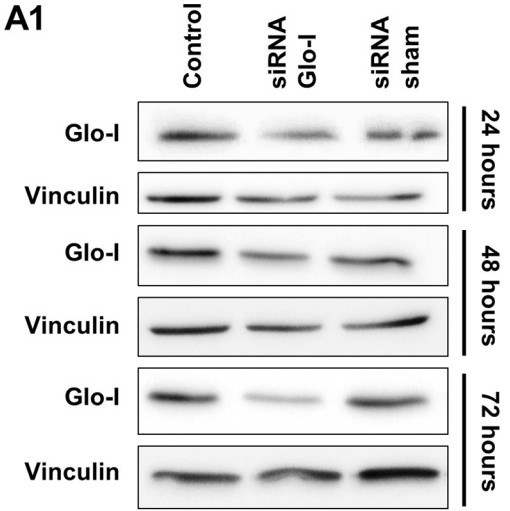

**A2**

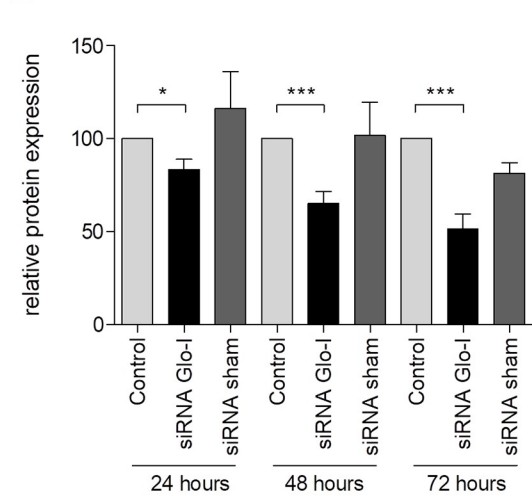

**B1**

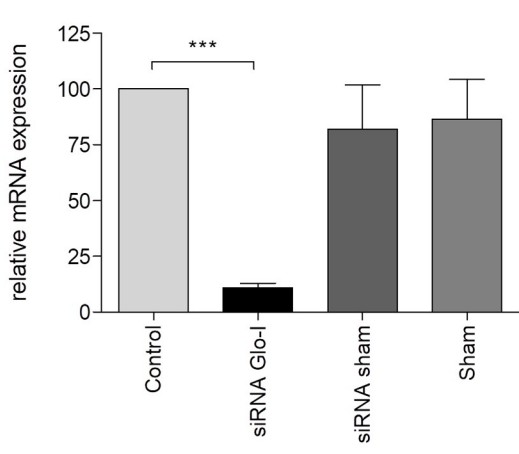

**B2**

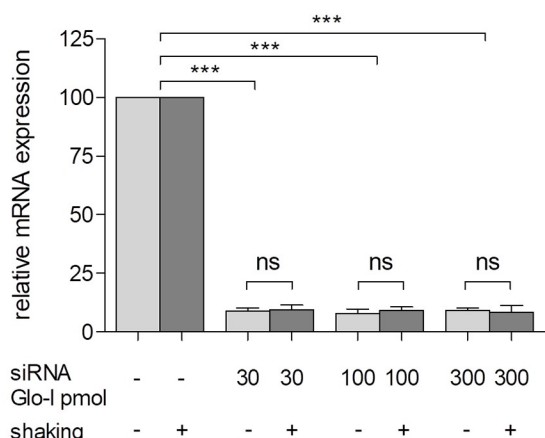

**B3**

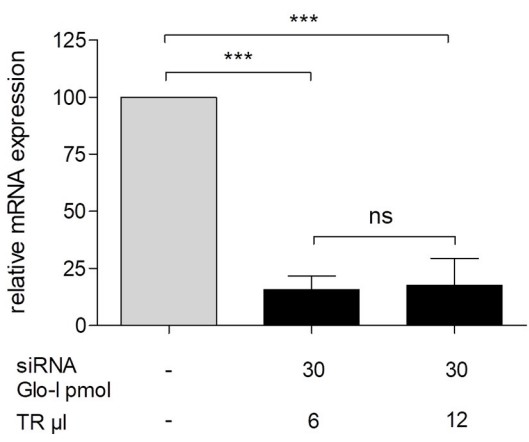

**Fig 5. Transient gene knockdown of AR42J cells by siRNA.** Efficacy of transfection of AR42J with siRNA with regard to the target Glo-I was analyzed. Cells were incubated with 100 nM dexamethasone for 48 hours and then transfected with Glo-I siRNA for 24 and up to 72 hours. WB analysis **(A1)** indicated reduction of Glo-I on protein levels, quantification showed significant reduction after 24 hours and highest reduction after 72 hours of incubation **(A2)**. **B1-B3**, analysis on mRNA-levels in qRT-PCR. Cells were incubated with Glo-I siRNA for 24 hours. Results showed a significant reduction of Glo-I expression compared to controls and sham treatment **(B1)**. Low concentrations of siRNA (30 pmol in 2 ml volume) were as effective as high amounts (300 pmol in 2 ml volume) and the additional use of the shaker did not optimize the transfection efficacy **(B2)**. Also, doubled amounts of transfection reagent did not affect the results **(B3)**. Glo-I: Glyoxalase-I. TR: transfection reagent, WB: Western Blot. Results are expressed as mean ± S.D. of three independent experiments. * P<0.05, *** P<0.001.

AR42J with higher passage grew more rapidly and showed both morphological and functional alterations.

The differentiation of AR42J to an acinar-like phenotype is mandatory for its use as an *in-vitro* model of acute CN-induced pancreatitis. Our data clearly show that the incubation of AR42J with dexa prior to CN stimulation is an essential step to enable amylase production and secretion in AR42J. Dexa led to a dramatic increase in the production of digestive enzymes and granula for secretion. Moreover, CN was able to induce secretion of amylase to supernatant only in dexa pre-treated cells. Although a substantial number of recently published studies did not mention dexa in the AR42J-model of AP (see Table 1), this step is compellingly required. We clearly illustrated that CN without dexa is not able to increase production nor secretion of amylase. Also, LPS was not able to induce amylase release, although it was reported [10,27,30,33]. A possible explanation for this contradistinction in our described method and results compared to the published protocols could be the fact that the use of dexa is self-evident when utilizing AR42J cells and therefore not mentioned explicitly. In this regard, two important points have to be considered. In the latter case, all the materials and methods sections would be imprecise and would not be in concordance to the principle of good laboratory research. Moreover, our experiences show that dexa cannot be added to cell culture permanently. Dexa led to reduced proliferation and, thus, should only be used for the differentiation of AR42J within the experiments. In addition, the proof of amylase secretion upon CN was not presented in all of the mentioned papers. This should also be taken into account when interpreting our data.

Next, we analyzed the use of inflammatory markers in the model of AR42J-induced pancreatitis. Several studies (see Table 1) described a CN-induced increase of TNF-α and IL-6, mostly by ELISA but also WB and qRT-PCR. Interestingly, we were not able to show any influence of CN on secretion of TNF-α or translation on mRNA levels. The same was true for IL-6. Nevertheless, analysis of TNF-α in protein lysates confirmed a reliable stimulation of TNF-α upon CN treatment. A possible explanation for this contradiction could be the use of different doses of CN. Indeed, some researchers used up to 1000 nM CN [12,51] but others described stimulation of TNF-α and IL-6 using doses of 10 nM CN [9,11,32,33,43,46,48,60]. Furthermore, recent studies used incubation times ranging between several minutes and 48 hours that might explain the discrepant results. In addition, we were able to show that AR42J should not be used after the passage of 35, as they reveal distinct functional and morphological properties. Eventually, the use of different passages of cells could have influenced the results published in the literature.

In contrast, we could clearly show that CN was able to induce the expression of p65 subunit of NF-κB. This is in line with published data (see Table 1) and therefore, shows that induction of NF-κB demonstrates a reliable parameter to analyze the inflammatory response of CN. In addition, we clearly showed that dexa did not influence the expression of NF-κB.

Transient transfection of cells is an important method to analyze the effects of gene silencing as well as overexpression and has been used as a standard procedure for several years.

**A**

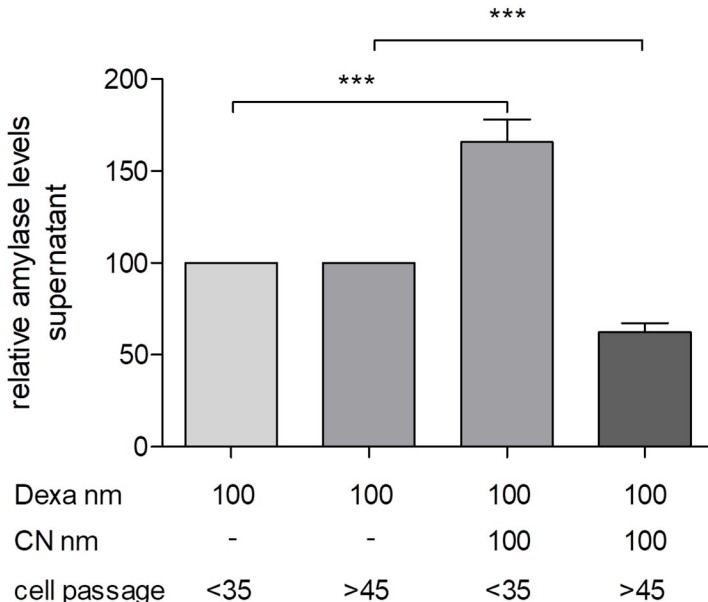

**B**

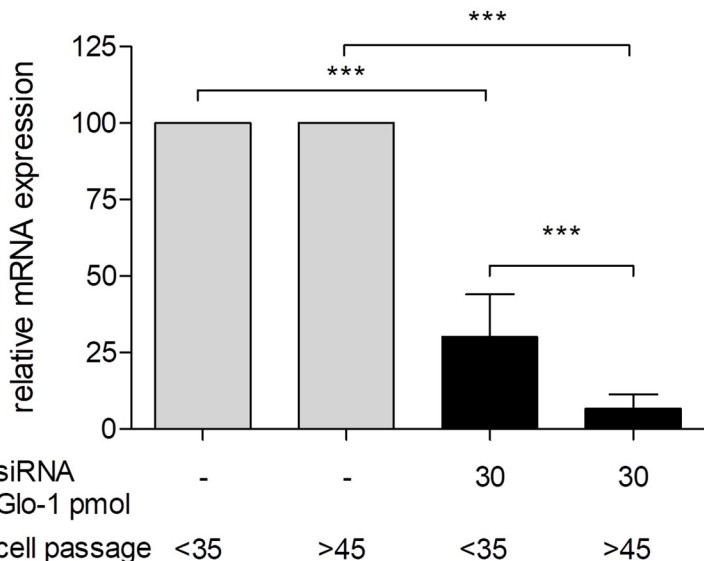

**Fig 6. Effect of AR42J cell passage on experimental setting.** The effect of cell processing on capability of amylase release (**A**) and transfection efficacy (**B**) was analyzed. AR42J below the passage of 35 were compared with cells beyond passage 45. **A**, AR42J up to passage 35 showed significant increase in amylase secretion upon cerulein (CN) stimulation but not AR42J beyond passage 45. Cells were incubated with dexamethasone for 48 hours and then stimulated with CN for additional 24 hours. Amylase release was measured by ELISA in supernatant. **B**, analysis of transfection efficacy with regard to Glo-I. More processed AR42J revealed higher efficacy in siRNA transfection indicated by reduced expression of Glo-I in qRT-PCR. CN: cerulein, Dexa: dexamethasone, Glo-I: Glyoxalase-I. Results are expressed as mean ± S.D. of three independent experiments. *** $P < 0.001$.

Transfection of AR42J was described in several publications but details to optimize transfections efficacy are lacking [30,31,34,39,40,42,45,48,51–53,59,60,62]. Thus, we analyzed the influence of different conditions to find an optimized transfection protocol. We could clearly show that a gene knockdown by siRNA was detectable in AR42J after 24 hours of incubation and was highest after 72 hours. Moreover, overexpression by plasmid-constructs resulted in significantly elevated protein expression after 6 hours and was highest after 12 hours of incubation. Thus, our protocol resulted in reliable silencing or overexpression and could be used as a standard in transfection of AR42J cells.

In conclusion, the model of CN-induced AP as well as the transient transfection for AR42J cells demonstrate reliable *in-vitro* methods but need specific conditions in order to obtain reproducible data.

## Supporting information

**S1 Fig.**
(PDF)

## Acknowledgments

We would like to thank Tiffany Schaumburg for her critical language editing.

## Author Contributions

**Conceptualization:** Marcus Hollenbach, Albrecht Hoffmeister.

**Data curation:** Sebastian Sonnenberg, Ines Sommerer, Jana Lorenz.

**Formal analysis:** Marcus Hollenbach, Sebastian Sonnenberg.

**Funding acquisition:** Albrecht Hoffmeister.

**Investigation:** Sebastian Sonnenberg, Ines Sommerer, Jana Lorenz.

**Methodology:** Sebastian Sonnenberg, Ines Sommerer, Jana Lorenz.

**Project administration:** Marcus Hollenbach, Albrecht Hoffmeister.

**Resources:** Albrecht Hoffmeister.

**Software:** Sebastian Sonnenberg, Ines Sommerer, Jana Lorenz.

**Supervision:** Marcus Hollenbach, Ines Sommerer, Jana Lorenz, Albrecht Hoffmeister.

**Validation:** Jana Lorenz.

**Visualization:** Sebastian Sonnenberg.

**Writing – original draft:** Marcus Hollenbach.

**Writing – review & editing:** Sebastian Sonnenberg, Ines Sommerer, Jana Lorenz, Albrecht Hoffmeister.

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
