## [Decision Letter · Decision Letter 0]

9 Jun 2020

PONE-D-20-13635

Pitfalls in AR42J-model of Cerulein-Induced Acute PancreatitisPitfalls in AR42J-model of Cerulein-Induced Acute Pancreatitis

PLOS ONE

Dear Dr. Hollenbach,

Thank you for submitting your manuscript to PLOS ONE. After careful consideration, we feel that it has merit but does not fully meet PLOS ONE’s publication criteria as it currently stands. Therefore, we invite you to submit a revised version of the manuscript that addresses the points raised during the review process.

Your paper has been evaluated by three experts in the field. Overall, the paper is well-written, there are only some minor points that need to be addressed. Novelty is not an issue at PLOS ONE, you do not need to deal with these comments.

We look forward to receiving your revised manuscript.

Kind regards,

Zoltán Rakonczay Jr., M.D., Ph.D., D.Sc.

Academic Editor

PLOS ONE

Journal Requirements:

2. At this time, we ask that you please provide full and detailed methodology of the ELISA experiments carried out in your study in the Methods section, to ensure that other researchers can replicate and reproduce your experiments. In addition, please revise the names of the antibodies in your Western Blot section to include the prefix "anti-" (e.g. anti-alpha-amylase).

3. Please report your cell concentrations in scientific notation.

6. Please amend either the title on the online submission form (via Edit Submission) or the title in the manuscript so that they are identical.

7. Please include your tables as part of your main manuscript and remove the individual files. Please note that supplementary tables should be uploaded as separate "supporting information" files.

Reviewers' comments:

Reviewer's Responses to Questions

**Comments to the Author**

1. Is the manuscript technically sound, and do the data support the conclusions?

Reviewer #1: Partly

Reviewer #2: Yes

Reviewer #3: Yes

2. Has the statistical analysis been performed appropriately and rigorously? 

Reviewer #1: Yes

Reviewer #2: Yes

Reviewer #3: Yes

3. Have the authors made all data underlying the findings in their manuscript fully available?

Reviewer #1: Yes

Reviewer #2: Yes

Reviewer #3: Yes

4. Is the manuscript presented in an intelligible fashion and written in standard English?

Reviewer #1: Yes

Reviewer #2: Yes

Reviewer #3: Yes

5. Review Comments to the Author

Reviewer #1: In this manuscript, the authors presented AR42J cells as a cell-culture model of cerulein (CER)-induced acute pancreatitis as well as the conditions for the transfection and induction of proinflammatory markers in CER-treated AR42J cells. Unfortunately, there is no novelty in the study, because it is well known that these amphicrine cells should be treated by dexamethasone that favors their differentiation toward the exocrine phenotype and leads to increased secretion of amylase in response to the secretagogue cholecystokinin. AR42J cells have been also widely used as an “in vitro” model for the transfection to study the exocrine pancreas and it was described in published protocols and reviews (Logsdon CD, Moessner J, Williams JA, Goldfine ID. Glucocorticoids increase amylase mRNA levels, secretory organelles, and secretion in pancreatic acinar AR42J cells. J Cell Biol. 100:1200-1208, 1985; Gonzalez A S-CP, Salido GM (2011) Culture of pancreatic AR42J cell for use as a model for acinar cell function. In: The Pancreapedia: Exocrine Pancreas Knowledge Base 2011; Christophe J. Pancreatic tumoral cell line AR42J: an amphicrine model. Am J Physiol 266: G963-971, 1994; Mareninova, O.A. Orabi, Abrahim I. Husain, Sohail Z. (2015). Experimental acute pancreatitis: In vitro models. In: The Pancreapedia: Exocrine Pancreas Knowledge Base 2015).

Reviewer #2: The article written by Hollenbach et al. provides a useful summary about the methodological difficulties that any researcher may face when working with AR42J cell line. The authors also created a guidance of the optimal experimental setup of this cell line that helps to follow good laboratory practice.

I have only a few reflections that should be clarified in the manuscript:

1. Abstract/ Results: “CN treatment resulted in increased TNS-alpha production but not secretion and did not influence IL-6.” The second part of this sentence should be corrected. Which characteristics of IL-6 is mentioned? Expression, IL-6 levels, production etc.?

2. Materials and methods / qRT-PCR: The details of cDNA construction should be mentioned here.

3. Materials and methods / Glo-I plasmid generation: Primer orientation should be indicated (5’ – 3’), and heat shock temperature of E.coli transformation should be mentioned.

4. Figure 2C. It is mentioned that LPS was not able to induce a secretion of amylase by AR42J. However, it seems that LPS treatment inhibited the increase of amylase levels after dexamethason pretreatment and CN treatment. According to the Figure2C amylase levels decreased if cells were treated by LPS and CN together compared to CN only treatment. Could you please explain this observation? Is it possible that this figure is inaccurate?

Reviewer #3: Hollenbach et al. present a methodological paper on the usability of AR42J cells for investigation of some aspects of the pathophysiology in acute pancreatitis. They show that caerulein can be used in combination with dexamethasone to induce amylase and cytokine secretion. They also investigated the expression of nf-κΒ and transfected cells with a plasmid containing glyoxalase-I or glyoxalase-I siRNA. The authors conclude that AR42J cells are a useful in-vitro model for caerulein induced acute pancreatitis when considering some experimental conditions.

The paper covers an important topic when investigating experimental pancreatitis. The methods are clearly presented and the results contain novelty.

There are some aspects the authors need to consider:

1.) In the images of figure 1 and figure 4, A1 and A3 the brightness of AR42J cells should be adjusted.

2.) The authors should mention in the introduction why the investigate Glyoxalase-I expression in AR42J cells.

3.) It should be mentioned in the results section (p. 11-12), which concentration of CN leads to the highest amylase secretion.

4.) The figures should be arranged in a numerical order starting with figure 1. Here the authors start with figure 6A (p. 11).

6. PLOS authors have the option to publish the peer review history of their article (what does this mean?). If published, this will include your full peer review and any attached files.

Reviewer #1: No

Reviewer #2: No

Reviewer #3: No

---

## [Author Response · Author response to Decision Letter 0]

24 Oct 2020

Reviewer #1: 

In this manuscript, the authors presented AR42J cells as a cell-culture model of cerulein (CER)-induced acute pancreatitis as well as the conditions for the transfection and induction of proinflammatory markers in CER-treated AR42J cells. Unfortunately, there is no novelty in the study, because it is well known that these amphicrine cells should be treated by dexamethasone that favors their differentiation toward the exocrine phenotype and leads to increased secretion of amylase in response to the secretagogue cholecystokinin. AR42J cells have been also widely used as an “in vitro” model for the transfection to study the exocrine pancreas and it was described in published protocols and reviews (Logsdon CD, Moessner J, Williams JA, Goldfine ID. Glucocorticoids increase amylase mRNA levels, secretory organelles, and secretion in pancreatic acinar AR42J cells. J Cell Biol. 100:1200-1208, 1985; Gonzalez A S-CP, Salido GM (2011) Culture of pancreatic AR42J cell for use as a model for acinar cell function. In: The Pancreapedia: Exocrine Pancreas Knowledge Base 2011; Christophe J. Pancreatic tumoral cell line AR42J: an amphicrine model. Am J Physiol 266: G963-971, 1994; Mareninova, O.A. Orabi, Abrahim I. Husain, Sohail Z. (2015). Experimental acute pancreatitis: In vitro models. In: The Pancreapedia: Exocrine Pancreas Knowledge Base 2015).

We thank the reviewer for this advice but clearly disagree with the reviewer’s opinion. Indeed, AR42J cells were used for decades as an in-vitro model for cerulein (CN)-induced pancreatitis. However, as demonstrated in detail in Table 1, the used conditions for that model dramatically vary between the published articles. Moreover, important details of the experimental setting often are missing. Our data therefore clearly describe the necessity of dexamethasone-treatment prior to CN-stimulation. In addition, different concentrations of CN and dexamethasone as well as incubation times were analyzed. Furthermore, we present an optimized protocol for successful overexpression and gene knockdown to maximize transfection efficacy. 

Reviewer #2: The article written by Hollenbach et al. provides a useful summary about the methodological difficulties that any researcher may face when working with AR42J cell line. The authors also created a guidance of the optimal experimental setup of this cell line that helps to follow good laboratory practice.

We thank the reviewer for the positive critiques and we hope our article will help researches to optimize their own AR42J experimental setting.

I have only a few reflections that should be clarified in the manuscript:

1. Abstract/ Results: “CN treatment resulted in increased TNS-alpha production but not secretion and did not influence IL-6.” The second part of this sentence should be corrected. Which characteristics of IL-6 is mentioned? Expression, IL-6 levels, production etc.?

We thank the reviewer for this advice and corrected the corresponding sentence.

2. Materials and methods / qRT-PCR: The details of cDNA construction should be mentioned here.

We used the Qiagen QuantiTect one stop qRT-PCR kit. This kit allows the easy and fast qRT-PCR without necessity of checking cDNA. RNA was used as described in the manufacturers’ manual. We updated the information in this section regarding the used PCR protocol.

3. Materials and methods / Glo-I plasmid generation: Primer orientation should be indicated (5’ – 3’), and heat shock temperature of E.coli transformation should be mentioned.

This section of material and methods was revised and the requested information provided.

4. Figure 2C. It is mentioned that LPS was not able to induce a secretion of amylase by AR42J. However, it seems that LPS treatment inhibited the increase of amylase levels after dexamethason pretreatment and CN treatment. According to the Figure2C amylase levels decreased if cells were treated by LPS and CN together compared to CN only treatment. Could you please explain this observation? Is it possible that this figure is inaccurate?

We are very thankful for this advice. Indeed, fig. 2C was incomplete. Our data showed that LPS treatment alone was not able to induce amylase secretion. In addition, co-treatment of LPS and CN did not further influence the release of amylase compared to CN alone. We corrected fig. 2C and clarified the statement in the manuscript.

Reviewer #3: Hollenbach et al. present a methodological paper on the usability of AR42J cells for investigation of some aspects of the pathophysiology in acute pancreatitis. They show that caerulein can be used in combination with dexamethasone to induce amylase and cytokine secretion. They also investigated the expression of nf-κΒ and transfected cells with a plasmid containing glyoxalase-I or glyoxalase-I siRNA. The authors conclude that AR42J cells are a useful in-vitro model for caerulein induced acute pancreatitis when considering some experimental conditions. The paper covers an important topic when investigating experimental pancreatitis. The methods are clearly presented and the results contain novelty.

We thank the reviewer for the positive evaluation of our manuscript. Indeed, we hope to provide evidence and guidance for researches that are faced with the AR42J-model of CN-induced pancreatitis. 

There are some aspects the authors need to consider:

1.) In the images of figure 1 and figure 4, A1 and A3 the brightness of AR42J cells should be adjusted.

We thank for this suggestion of optimized brightness of the corresponding images.

2.) The authors should mention in the introduction why the investigate Glyoxalase-I expression in AR42J cells.

The Glyoxalase-System demonstrates a main research interest for many years in the authors’ lab. Ongoing studies analyze the role of Glyoxalas-I in acute and chronic pancreatitis as well as pancreatic cancer in vivo and in vitro. This was highlighted in the introduction section. 

3.) It should be mentioned in the results section (p. 11-12), which concentration of CN leads to the highest amylase secretion.

It was now stated at results section that 100nM CN was used for the following experiments as this concentration induced highest amylase release.

4.) The figures should be arranged in a numerical order starting with figure 1. Here the authors start with figure 6A (p. 11).

We agree with the reviewer’s suggestion and referred in the corresponding paragraph to sections below. Now the figures order follows the results presentation in the manuscript.

---

## [Decision Letter · Decision Letter 1]

9 Nov 2020

Pitfalls in AR42J-model of Cerulein-Induced Acute Pancreatitis

PONE-D-20-13635R1

Dear Dr. Hollenbach,

We’re pleased to inform you that your manuscript has been judged scientifically suitable for publication and will be formally accepted for publication once it meets all outstanding technical requirements.

Kind regards,

Zoltán Rakonczay Jr., M.D., Ph.D., D.Sc.

Academic Editor

PLOS ONE

Additional Editor Comments (optional):

Reviewers' comments:

Reviewer's Responses to Questions

**Comments to the Author**

1. If the authors have adequately addressed your comments raised in a previous round of review and you feel that this manuscript is now acceptable for publication, you may indicate that here to bypass the “Comments to the Author” section, enter your conflict of interest statement in the “Confidential to Editor” section, and submit your "Accept" recommendation.

Reviewer #1: All comments have been addressed

Reviewer #2: All comments have been addressed

Reviewer #3: All comments have been addressed

2. Is the manuscript technically sound, and do the data support the conclusions?

Reviewer #1: Yes

Reviewer #2: Yes

Reviewer #3: Yes

3. Has the statistical analysis been performed appropriately and rigorously? 

Reviewer #1: Yes

Reviewer #2: Yes

Reviewer #3: N/A

4. Have the authors made all data underlying the findings in their manuscript fully available?

Reviewer #1: Yes

Reviewer #2: Yes

Reviewer #3: Yes

5. Is the manuscript presented in an intelligible fashion and written in standard English?

Reviewer #1: Yes

Reviewer #2: Yes

Reviewer #3: Yes

6. Review Comments to the Author

Reviewer #1: (No Response)

Reviewer #2: All of my questions were answered and the revised version of the manuscript was corrected accordingly.

Reviewer #3: The authors have clarified the reviewers’ queries and the quality of the manuscript could be improved.

7. PLOS authors have the option to publish the peer review history of their article (what does this mean?). If published, this will include your full peer review and any attached files.

Reviewer #1: No

Reviewer #2: No

Reviewer #3: No

---

## [Editor Report · Acceptance letter]

26 Nov 2020

PONE-D-20-13635R1 

Pitfalls in AR42J-model of Cerulein-Induced Acute Pancreatitis 

Dear Dr. Hollenbach:

I'm pleased to inform you that your manuscript has been deemed suitable for publication in PLOS ONE. Congratulations! Your manuscript is now with our production department. 

Kind regards, 

on behalf of

Dr. Zoltán Rakonczay Jr. 

Academic Editor

PLOS ONE